# WHAT MAKES A GOOD TIME-SERIES FORECASTING MODEL? A CAUSAL PERSPECTIVE

## ABSTRACT

Generalization is a long-standing challenge in multivariate time series forecasting (MTSF) tasks. Current approaches typically assume correlations among all variables. Consequently, every variable is incorporated into the training process for prediction tasks. From a causal perspective, this reliance on correlated variables can compromise the model's generalization. To address this, we aim to explore the role of causal relationships in enhancing the generalization of multivariate time series models. We examine how graphical causal models, through conditional independence constraints, can narrow down the hypothesis space, thereby improving generalization. Building on this foundation, we propose a novel causality-based MTSF algorithm **CAusal Informed Transformer** (CAIFormer). We first construct a Directed Acyclic Graph (DAG) among variables using causal discovery. Then we build the forecasting model by constructing the Markov boundary informed by the DAG. Empirical evaluations on benchmark datasets demonstrate that our method surpasses traditional approaches in predictive accuracy. Additionally, we present the Markov boundaries derived for these datasets, underscoring the practical applicability of our causality-driven framework in MTSF.

## 1 INTRODUCTION

Multivariate Time Series Forecasting (MTSF) is a fundamental problem in various fields, including energy consumption (Bilal et al., 2022), economic planning (Hidalgo, 2009), weather prediction (Duchon & Hale, 2012), and traffic forecasting (Li et al., 2015). It involves predicting future values of multiple interrelated variables based on their historical data (Box et al., 2015). With the advent of deep learning techniques (LeCun et al., 2015), numerous methods have been proposed to tackle MTSF tasks (Zhu et al., 2024; Hu & Xiao, 2022; Bai et al., 2018; Wen et al., 2023; Guo et al., 2023). Although these methods have achieved remarkable progress, improving their generalization ability remains a critical challenge. There are countless models that can achieve low empirical risk but may not generalize well to unseen data. According to prior works in Probably Approximately Correct (PAC) learning theory (Vapnik & Chervonenkis, 1971), without proper regularization of the model hypothesis space, models are prone to overfitting, leading to higher generalization risk (Mohri et al., 2018; Kuznetsov & Mohri, 2014).

An essential characteristic of MTSF is that the future behavior of each variable depends not only on its own historical data but also on the historical data of other variables. For instance, when predicting precipitation, changes in atmospheric pressure provide valuable information alongside historical precipitation data (Wilks, 2011). Consequently, existing methods often incorporate all available variables as inputs when forecasting the future sequence of a particular variable (Bai et al., 2018; Liu et al., 2023a; Zhang et al., 2024b; Zhan et al., 2023). However, indiscriminately including all variables may not always be the most effective strategy. From a causal inference perspective, the relationships among these variables can be intricate: for a specific variable, some variables may be causes, some may be effects, and some may be independent (Pearl, 2009; Glymour et al., 2016). By explicitly considering these causal relationships during model construction, we can leverage them to constrain the hypothesis space of the model, potentially improving generalization performance.

To investigate how causal relationships affect generalization in MTSF problems, we follow prior works by defining causal relationships using conditional independence (Dawid, 1979; Pearl & Paz, 2022; Pearl, 2009). We can conceptualize multivariate time series as a weighted representation of

all random variables. To simplify this representation, we aim to identify a maximal set of linearly independent variables, thereby uncovering the essential features that influence the evolution of the series. Instead of merely learning an effective representation from the training data, we strive for the model to possess the capability to identify the maximal set of linearly independent variables across diverse scenarios. The subset of random variables meeting these conditions is equivalent to the Markov boundary (Pearl, 1988; Statnikov et al., 2013a). Upon constructing the Markov boundary, we identify that collider structures in the Markov boundary introduce additional conditional independencies, which are frequently neglected by most MTSF methods. Furthermore, in our theoretical analysis, we examine the impact of collider structures on MTSF tasks and demonstrate that enforcing these conditional independencies can narrow down the hypothesis space of the forecasting model. This constraint can effectively reduce the generalization error theoretically, thereby enhancing the generalization performance of the model.

Based on our theoretical conclusions, we propose a novel algorithm named **CAusal Informed Transformer** (CAIFormer). Specifically, we first employ causal discovery algorithms to construct a DAG that captures the relationships among variables in an MTSF task. We then develop an algorithm to extract the Markov boundary for all variables in the DAG. Subsequently, we integrate these insights into a Transformer-based forecasting model by constraining each variable's attention module to focus exclusively on the variables within its Markov boundary. This approach effectively leverages causal relationships to enhance the model's generalization performance.

Our proposed CAIFormer achieves superior performance of the SOTA methods on a series of benchmarks. Additionally, we conduct a series of ablation studies to assess the impact of different causal discovery algorithms and hyperparameter settings on the effectiveness of CAIFormer. Furthermore, we include DAGs of various MTSF datasets in the Appendix B, aiming to inspire future research. Our contributions can be summarized as follows:

- We explore the causal relationships among variables and discover that the Markov boundary is the sufficient and necessary subset of all variables in forecasting tasks.

- We demonstrate that the collider structure within the Markov boundary contains additional conditional independence, which enables us to constrain the hypothesis space of the forecasting model, ultimately improving the generalization ability of the model.

- We propose a novel algorithm, CAusal Informed Transformer (CAIFormer), which integrates causal relationships into a Transformer-based model. CAIFormer constrains each variable's attention module to focus solely on the variables within its Markov boundary.

- We demonstrate that CAIFormer outperforms SOTA methods on multiple benchmarks. The ablation studies showcase the correctness of our proposed method.

## 2 RELATED WORK

**Multivariate time series forecasting** aims to predict future values of multiple, potentially interrelated variables based on historical data (Box et al., 2015; Lim & Zohren, 2021; Zhang et al., 2024a). Traditional MTSF methods often employ autoregressive models (Box et al., 2015), exponential smoothing (Gardner Jr, 1985; Winters, 1960), or structural time series models (Harvey, 1990). With the advancement of deep learning, various methods including CNNs (Zhan et al., 2023; Bai et al., 2018), RNN (Hewamalage et al., 2021; Tang et al., 2021), and MLP-based (Zeng et al., 2023; Li et al., 2023; Zhang et al., 2024b) methods were proposed. In addition, there are Transformer-based models that utilize a self-attention mechanism to compute relationships between variables, while applying causal inference to restrict the calculations to variables with causal connections.

**Generalization Analysis in Time Series Forecasting.** The generalization problem refers to a model's ability to maintain performance on unseen data (Mohri et al., 2018). Given a finite number of training samples, the Probably Approximately Correct (PAC) learning framework ensures the model's generalization error remains below a predetermined threshold with high probability (Valiant, 1984). The threshold, which is generally called generalization bound, depends on the complexity of the model's hypothesis space (Koltchinskii, 2001; Vapnik & Chervonenkis, 1971). In time series forecasting, early works assume stationarity and suitable mixing conditions (Doukhan & Doukhan, 1994). For instance, Yu Yu (1994) established VC-dimension bounds for binary classification under the assumptions of stationarity and $\beta$-mixing. (Kuznetsov & Mohri, 2015) proposed

generalization bounds based on sequential Rademacher complexity (Rakhlin et al., 2010). In this paper, we extract the Markov boundary which is derived from causal relationships between variables, enabling explicit constraints on the hypothesis in MTSF.

**Causal Inference and Causal Discovery.** Causal inference seeks to deduce causal relationships among variables from observational data (Glymour et al., 2016; Pearl, 2009), typically represented by Directed Acyclic Graphs (DAGs) (Lauritzen & Wermuth, 1989). The Inductive Causation (IC) algorithm, introduced by (Verma & Pearl, 1990), constructs DAGs using conditional independence tests (CITs) to identify dependencies between variables. Based on this, (Spirtes & Glymour, 1991) developed the Peter-Clark (PC) algorithm, which has been refined to reduce the computational complexity(Spirtes et al., 2001; Spirtes, 2001). In time series data, causal discovery methods such as tsFCI (Entner & Hoyer, 2010) apply the Fast Causal Inference (FCI) algorithm, while Granger causality (Granger, 1969) explores temporal cause-effect relationships. Recent works have integrated causal knowledge to enhance forecasting models: (Li et al., 2021) proposed a hidden causal Markov model to reduce spurious correlations, and (Liu et al., 2023a) used proxy variables to uncover complete causal structures. Unlike previous approaches, we leverage DAGs from causal discovery to constrain model parameters, significantly improving the generalization in MTSF.

## 3 PRELIMINARY

In this section, we first introduce the problem setting of MTSF (Section 3.1). Next, we provide the background in causality with multiple definitions (Section 3.2).

### 3.1 MULTIVARIATE TIME-SERIES FORECASTING

Multivariate time series forecasting (MTSF) is a sequence-to-sequence problem. Let $X_{1:T} = \{x_{1:T}^1, x_{1:T}^2, \ldots, x_{1:T}^D\} \in \mathbb{R}^{T \times D}$ represent the historical sequence with $T$ time steps and $D$ variables. At any timestamp $t$, the state of the variables is represented as $X_t = \{x_t^1, x_t^2, \ldots, x_t^D\} \in \mathbb{R}^D$. MTSF aims to predict the future sequence $X_{T+1:T+S} = \{x_{T+1:T+S}^1, x_{T+1:T+S}^2, \ldots, x_{T+1:T+S}^D\} \in \mathbb{R}^{S \times D}$ by maximizing the following conditional distribution:

$$P(X_{T+1:T+S} \mid X_{1:T}; \theta), \tag{1}$$

where $\theta$ represents the learnable parameters. Given a training dataset $D_{\text{train}} = \{(X_{1:T}^i, X_{T+1:T+S}^i)\}_{i=1}^m$, the learning objective of MTSF can be formalized as learning a parameterized function $\hat{f}_\theta$ that estimates the optimal predictor $f^*$, where $f^*(X_{1:T}) = X_{T+1:T+S}$, from the hypothesis space $\mathcal{F}$ by solving the empirical risk minimization problem:

$$\hat{f}_\theta = \arg\min_{f \in \mathcal{F}} \sum_{i=1}^m L(X_{T+1:T+S}^i, f(X_{1:T}^i)), \tag{2}$$

where $L$ denotes the loss function. Note that in Equation 1, without additional constraints, the future value of each variable depends on all other variables.

### 3.2 BACKGROUND IN CAUSALITY

Causality examines how changes in one random variable influence another based on their probabilistic relationships Pearl (2009). One of the core concepts of causality is the conditional independence, which we provide the definition as follows:

**Definition 1 (Conditional Independence Dawid (1979))** *Let $V = \{V_1, V_2, ...\}$ be a finite set of variables, $P(\cdot)$ be a joint probability function over the variables in $V$, and $X$, $Y$, $Z$ stand for any three subsets of variables in $V$. Then, $X$ and $Y$ are said to be conditionally independent given $Z$ if*

$$P(X|Y, Z) = P(X|Z) \text{ whenever } P(Y, Z) > 0. \tag{3}$$

*In words, learning the value of $Y$ does not provide additional information about $X$, once we know $Z$. We will use the $X \perp\!\!\!\perp Y|Z$ to represent the conditional independence of $X$ and $Y$ given $Z$.*

Conditional independence relationships among variables form the basis of causal graph models. In these models, a Directed Acyclic Graph (DAG), denoted as $G = (V, E)$, is typically used to represent the relationships between variables, where the node set $V = \{V_1, V_2, \ldots\}$ corresponds to random variables, and the edge set $E = \{(V_1, V_2), (V_2, V_3), \ldots\}$ represents causal relationships. Causal graph models are built upon three fundamental structures: Chain, Fork, and Collider. Any model containing at least three variables incorporates these key structures. Figure 1 illustrates examples of these structures.

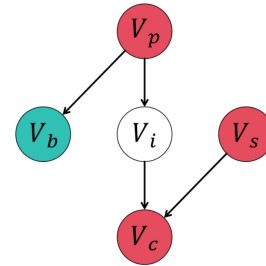

Figure 1: An example of the Causal Graph Model. The Markov boundary of $V_i$ is highlighted in red, and the variable outside the Markov boundary is highlighted in blue.

**Definition 2 (Chain)** *A chain $V_p \rightarrow V_i \rightarrow V_c$ is a graphical structure involving three variables $V_p$, $V_i$, and $V_c$ in graph G, where $V_p$ has a directed edge to $V_i$ and $V_i$ has a directed edge to $V_c$. Here, $V_p$ causally influences $V_i$, and $V_i$ causally influences $V_c$, making $V_i$ a mediator.*

In a chain structure, $V_p$ and $V_c$ are conditionally independent given $V_i$, formally, $V_p \perp\!\!\!\perp V_c \mid V_i$. This is because once the mediator $V_i$ is accounted for, knowing $V_p$ provides no additional information about $V_c$ beyond what is already conveyed through $V_i$.

**Definition 3 (Fork)** *A fork $V_b \leftarrow V_p \rightarrow V_i$ is a graphical structure involving $V_b$, $V_p$, and $V_i$, where $V_p$ is a common parent of both $V_b$ and $V_i$. Here, $V_p$ causally influences $V_b$ and $V_i$.*

Here, $V_b$ and $V_i$ are conditionally independent given the common parent $V_p$. It means that once $V_p$ is known, $V_b$ provides no additional information about $V_i$, and vice versa, i.e., $V_b \perp\!\!\!\perp V_i \mid V_p$.

**Definition 4 (Collider/V-Structure)** *A collider, also known as a V-structure, $V_i \rightarrow V_c \leftarrow V_s$, is a graphical structure involving three variables $V_i$, $V_c$, and $V_s$, where $V_c$ is a common child of both $V_i$ and $V_s$, $V_i$ and $V_s$ are not directly connected. Here, $V_i$ and $V_s$ causally influence $V_c$*

In a collider or V-structure, $V_i$ and $V_s$ are marginally independent; knowing $V_i$ does not provide information about $V_s$ and vice versa. However, when conditioning on the collider $V_c$, this independence is broken, making $V_i$ and $V_s$ dependent. Formally, $V_i \perp\!\!\!\perp V_s$ and $V_i \not\!\perp\!\!\!\perp V_s \mid V_c$.

These conditional independence relationships are fundamental for understanding the dependencies and independencies implied by a causal graph, thereby facilitating tasks such as causal discovery and inference in multivariate time series forecasting.

## 4 THEORETICAL ANALYSIS

In this section, we start by reviewing the concept of Markov boundaries and how it is used for MTSF. We then show that incorporating probabilistic inductive bias from a collider structure into an MTSF problem provides guarantees of improved generalization error. For the sake of clarity, our exposition focuses on the simple causal structure in Figure 1.

### 4.1 MARKOV BOUNDARY AND CONDITIONAL INDEPENDENCE

Without loss of generality, multivariate time series forecasting can be regarded as an auto-regressive problem (Box et al., 2015). That is, suppose that there are $k$ random variables contained in a multivariate time series $Y$. From the perspective of linear algebra, the series $Y$ can be represented as a weighted sum of all random variables $X = \{X_i\}_{i=1}^{k}$. Due to the correlation between these random variables, there must be a subset within the set of variables such that the time series can be represented by, and only by, all the variables in that subset. In other words, there exists a maximal linearly independent group $X^* = \{X_m\}_{m=1}^{l}, (l < k)$ such that conditional independence $Y \perp\!\!\!\perp X \setminus X^* | X^*$ is maintained. Therefore, we can discard $X \setminus X^*$ from the total set without any loss of probabilistic

information for auto-regression. The set $X^*$ that satisfies conditional independence is also known as the Markov boundary (Statnikov et al., 2013b) of $Y$.

Moreover, the presence of collider structures within the Markov boundary provides additional independence relationships, thus improving the auto-regression, which is essentially a conditional distribution with the form of $P(Y|X)$. The following proposition shows that the presence of a collider is not only a sufficient condition but also necessary.

**Proposition 1** *(Koller & Friedman, 2009) Suppose that the Markov boundary of $Y$ is $X^*$, then $X^*$ contains a collider if and only if there exist $X_i \in X^*$ and $\tilde{X} \subset X^*$ such that $Y \perp\!\!\!\perp X_i | \tilde{X}$.*

For the sake of clarity, we discuss how conditional independence helps generalize under the collider structure in Figure 1. We claim that this simplification does not harm the generality of our work.

### 4.2 Hypothesis and Generalization under Markov boundary

Let $V_c, V_s, V_i$ be random variables following the collider structure in Figure 1. Under the auto-regression problem with squared loss, the optimal regressor is given by the following equation:

$$f^*(v_c, v_s) = \mathbb{E}[V_i | V_c = v_c, V_s = v_s]. \tag{4}$$

Here, the lowercase represents the values of $V_c, V_s$, respectively. With the independence relationship $V_i \perp\!\!\!\perp V_s$ given by the collider, we have:

$$\mathbb{E}[f^*(V_c, V_s)|V_s] = \mathbb{E}[\mathbb{E}[V_i|V_c, V_s]|V_s] = \mathbb{E}[V_i|V_s] = \mathbb{E}[V_i], \tag{5}$$

where the second equal comes from the tower property of the conditional expectation. Without loss of generality, we assume that $\mathbb{E}[V_i] = 0$. Hence, the optimal regressor lies in the subspace of functions with zero conditional expectation of $V_s$. To ensure accurate estimation, the function $\hat{f}$ lies within the same subspace where functions satisfy the zero conditional expectation constraint below.

$$\hat{f} \in \{f \in \mathcal{F} | \mathbb{E}[f(V_c, V_s)|V_s] = 0\}. \tag{6}$$

Starting with the general case of square-integrable functions, we propose to show how such a constraint hypothesis benefits generalization. Let $L^2(V)$ denote the space of square-integrable functions with respect to the probability measure induced by $V$ and suppose $\mathcal{F} = L^2(V)$. Let $E : L^2(V) \to L^2(V)$ denote the conditional expectation operator defined by:

$$Ef(v_c, v_s) = \mathbb{E}[f(V_c, V_s)|V_s]. \tag{7}$$

The operator $E$ classically defines an orthogonal projection over the subspace of $V_s$-measurable functions. $L^2(V)$ thus orthogonally decomposes into its projection, denoted $Range(E)$, and its null-space, denoted $Ker(E)$, as follows:

$$L^2(V) = Range(E) \oplus Ker(E). \tag{8}$$

Recall the constraint in Equation 6, we want to find the optimal regressor satisfies $\hat{f} \in Ker(E)$. For convenience, denote by $M = Id - E$ the orthogonal projection onto $Ker(E)$, then $\mathcal{F} = Range(M)$ is our hypothesis space. However, in practice, it may be hard to directly constrain the hypothesis space to be $Range(M)$, but the solution to the auto-regression problem with the squared loss function can orthogonally decompose within $L^2(V)$ as follows:

$$\hat{f} = M\hat{f} + E\hat{f}. \tag{9}$$

We emphasize that discarding $E\hat{f}$ can always yield generalization benefits.

**Theorem 1** *Let $f \in L^2(V)$ be any regressor from our hypothesis space. We have*

$$\Delta(f, Mf) = \|Ef\|^2_{L^2(V)}. \tag{10}$$

The generalization gap is always greater than zero. Hence, for any given regressor $\hat{f}$, we can always improve its test performance by projecting it onto $Range(P)$. See the proof in Appendix A.

# 5 THE PROPOSED METHOD

In this section, we present the framework of the CAIFormer method. We first extract causal relationships between variables from the dataset using the constraint-based Causal Discovery algorithm. Second, we find the Markov boundary for every variable based on the casual DAG. Next, we use the Transformer as the backbone and impose constraints on self-attention based on the Markov boundary. Finally, we constrain the hypothesis space through a specific structure.

## 5.1 CAUSAL DISCOVERY

In this section, we aim to explore the relationship between different variables in the dataset. The dataset comprises a set of random variables $V = \{V_1, V_2, ..., V_D\}$, where rows correspond to timestamps, and columns represent different variables.

To identify these relationships, we apply the Peter-Clark (PC) algorithm, a constraint-based Causal Discovery method, which reconstructs a Partially Directed Acyclic Graph (PDAG) by identifying conditional independencies. The PDAG consists of both directed and undirected edges. The directed edges denote definite causal relationships, while undirected edges reflect no fixed direction in causal relationships.

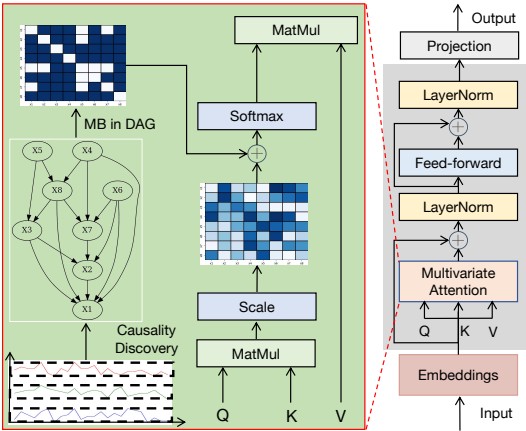

Figure 2: The framework of CAIFormer.

The PC algorithm systematically searches for separating sets $S_{ab}$, removing edges from the complete graph when separation is found. This way starts with empty sets $S_{ab}$ (cardinality 0), then cardinality 1, and so on, edges are recursively removed from a complete graph as soon as separation is found and has polynomial time in graphs of finite degree because at every stage the search for $a$ separating set $S_{ab}$ can be limited to nodes that are adjacent to $a$ and $b$. We prevent the details of the PC algorithm and visualize the resulting DAGs across different datasets in Appendix B.

Overall, by applying the causal discovery algorithm, we obtain a PDAG representing the causal relationships between variables in the dataset. Meanwhile, we get the $D \times D$ adjacency matrix, where $W_{adjm}[i][j] = 0$ means no edge between variable $V_i$, otherwise, there is an edge.

## 5.2 MARKOV BOUNDARY IN DAG

In Section 5.1, we utilized the causal discovery algorithm to extract the causal DAG and its adjacency matrix from the dataset. According to the analysis in Section 4.1, causal relationships for variable $V_i$ exist solely with variables within its Markov boundary. Thus, we aim to identify this boundary for every variable based on the causal graph and adjacency matrix. As illustrated in Figure 1, the process for determining the Markov boundary of feature $V_i$ involves two steps.

First, we identify the set of features $S_1^i$ that are dependent of $V_i$:

$$S_1^i = \{V_j | P(V_i) \neq P(V_i | V_j), V_j \in V\} \tag{11}$$

These features are represented in the DAG as either parent nodes (e.g., $V_p$) or child nodes (e.g., $V_c$). Parent nodes $V_p$ have directed edges towards $V_i$, while child nodes have edges directed from $V_i$. In the adjacency matrix, we identify the set of features connected to the $V_i$ node:

$$S_1^i = \{V_j | W_{adjm}[V_i][V_j] \neq 0, V_j \in \{1, 2, \cdots, n\}\} \tag{12}$$

In the adjacency matrix $W_{adjm}$, 1 signifies incoming edges and -1 denotes outgoing edges.

Next, we determine the set $S_2^i$ of features that remain not independent of $V_i$ given $S_1^i$:

$$S_2^i = \{V_j | (V_i \not\perp\!\!\!\perp V_j | S_1^i), V_j \in V \setminus S_1^i\} \tag{13}$$

Referring to Proposition 1, the elements of $S_2^i$ correspond to collider structures. Therefore, in the adjacency list, we locate all $V_j$ that share common child nodes with $V_i$:

$$S_2^i = \{V_j | \exists V_k, W_{adjm}[V_i][V_k] = adjm[V_j][V_k] = 1, V_j \in \{1, 2, \cdots, n\}\} \tag{14}$$

which mean $V_i$ have directed edges towards $V_k$ and $V_k$ have edges directed from $V_j$.

Finally, we combine $S_1^i$ and $S_2^i$ to obtain $S_{Mb}^i = S_1^i \cup S_2^i$, representing the Markov boundary of feature $V_i$. Using the same operation, we obtain the Markov boundary set of all variables in $V$ and generate Variable Attention Mask($V_{mask}$):

$$V_{mask}[i][j] = \begin{cases} 1 & \text{if } V_j \in S_{Mb}^i \\ 0 & \text{if } V_j \notin S_{Mb}^i \end{cases}, \quad \forall i \in (1, 2, \cdots, n), \forall j \in (1, 2, \cdots, n) \tag{15}$$

We visualize the mask for every dataset in Figure 3.

### 5.3 TRANSFORMER WITH VARIABLES MASK

Based on the above, we have got $V_{mask}$, where each element $V_{mask}[i][j]$ indicates relationship between $V_i$ and $V_j$. This determines whether $V_j$ should be considered when predicting $V_i$. To take advantage of the relationships between variables, we integrate the Transformer struct as backbone.

Self-Attention of Transformer captures the relationships between different tokens of input sequence by using each input vector as its own query, key, and value. Specifically, it begins with the input sequence represented as a matrix of size $T \times D$, where $T$ is the time steps and $D$ variates. From this input matrix, three matrices are generated through learned linear transformations: queries $Q = X \cdot W_Q$, keys $K = X \cdot W_K$, and values $V = X \cdot W_V$, where $W_Q, W_K$, and $W_V$ are weight matrices.

Next, calculate the dot product of each query with all keys, resulting in a matrix of similarity scores. To prevent the dot products from becoming excessively large, these scores are scaled by the square root of the dimension of the keys $d_k$. The scaled scores are then passed through the softmax function to produce attention weights, which sum to one, ensuring a probabilistic interpretation.

Finally, these attention weights are applied to the value matrix $V$ to obtain the output vector. This output reflects the contextualized representation of each input element, allowing the model to focus on relevant parts of the sequence dynamically. The overall self-attention calculation can be summarized as follows:

$$Attention(Q, K, V) = softmax(\frac{QK^T}{\sqrt{d_k}})V \tag{16}$$

The similarity score of different tokens is analogous to the weight matrix in linear models for feature selection, such as:

$$y = \sum_{i=1}^{n} w_i x_i, \tag{17}$$

where $W = \{w_1, w_2, \cdots, w_n\}$ play a role similar to that of the similarity score. In linear models, if a variable $x_i$ is independent of target $y$, as:

$$P(y|X) = P(y|X \setminus x_i), \tag{18}$$

where $X = \{x_1, x_2, \cdots, x_n\}$, then $x_i$ can be discarder in the prediction of $y$.

Similarly, when we rely on self-attention to compute the similarity score of other variables for $V_i$, we can discard independent variables of $V_i$. Based on the Markov boundaries obtained in Section 4.1, we can identify the causal relationships between each variable and other variables. Therefore, we impose constraints on self-attention to focus on the causal relationships among variables and only consider those variables within the Markov boundary. Specifically, our approach is as follows.

From Section 5.2, we derive a Variable Attention Mask, where each row indicates variables included in the Markov boundary of the current variable. We apply the mask to the similarity scores, where each variable acts as a token, representing the relationships between different variables. After applying the mask, for a specific variable $V_i$, we set the similarity scores with variables outside its Markov boundary to zero, thus avoiding irrelevant correlations.

Table 1: Multivariate time series forecasting results with prediction lengths $S \in \{96, 192, 336, 720\}$ and fixed lookback length $T = 96$. The best Forecasting results in **bold** and the second underlined. The lower MSE/MAE indicates the more accurate prediction result.

| Models | | CAIFormer | | iTransformer | | Crossformer | | TiDE | | TimesNet | | DLinear | | FEDformer | | Autoformer | |
|---|---|---|---|---|---|---|---|---|---|---|---|---|---|---|---|---|---|
| Metric | | MSE | MAE | MSE | MAE | MSE | MAE | MSE | MAE | MSE | MAE | MSE | MAE | MSE | MAE | MSE | MAE |
| ETTm1 | 96 | **0.327** | **0.364** | 0.334 | 0.368 | 0.404 | 0.426 | 0.364 | 0.387 | 0.338 | 0.375 | 0.345 | 0.372 | 0.379 | 0.419 | 0.505 | 0.475 |
| | 192 | **0.369** | **0.387** | 0.377 | 0.391 | 0.450 | 0.451 | 0.398 | 0.404 | 0.374 | 0.387 | 0.380 | 0.389 | 0.426 | 0.441 | 0.553 | 0.496 |
| | 336 | **0.411** | **0.412** | 0.426 | 0.420 | 0.532 | 0.515 | 0.428 | 0.425 | 0.410 | 0.411 | 0.413 | 0.413 | 0.445 | 0.459 | 0.621 | 0.537 |
| | 720 | 0.479 | **0.447** | 0.491 | 0.459 | 0.666 | 0.589 | 0.487 | 0.461 | 0.478 | 0.450 | 0.474 | 0.453 | 0.543 | 0.490 | 0.671 | 0.561 |
| | Avg | **0.396** | **0.402** | 0.407 | 0.410 | 0.513 | 0.496 | 0.419 | 0.419 | 0.400 | 0.406 | 0.403 | 0.407 | 0.448 | 0.452 | 0.588 | 0.517 |
| ETTm2 | 96 | **0.176** | **0.259** | 0.180 | 0.264 | 0.287 | 0.366 | 0.207 | 0.305 | 0.187 | 0.267 | 0.193 | 0.292 | 0.203 | 0.287 | 0.255 | 0.339 |
| | 192 | **0.245** | **0.304** | 0.250 | 0.309 | 0.414 | 0.492 | 0.290 | 0.364 | 0.249 | 0.309 | 0.284 | 0.362 | 0.269 | 0.328 | 0.281 | 0.340 |
| | 336 | **0.303** | **0.345** | 0.311 | 0.348 | 0.597 | 0.542 | 0.377 | 0.422 | 0.321 | 0.351 | 0.369 | 0.427 | 0.325 | 0.366 | 0.339 | 0.372 |
| | 720 | **0.405** | **0.401** | 0.412 | 0.407 | 1.730 | 1.042 | 0.558 | 0.524 | 0.408 | 0.403 | 0.554 | 0.522 | 0.421 | 0.415 | 0.433 | 0.432 |
| | Avg | **0.282** | **0.327** | 0.288 | 0.332 | 0.757 | 0.610 | 0.358 | 0.404 | 0.291 | 0.333 | 0.350 | 0.401 | 0.305 | 0.349 | 0.327 | 0.371 |
| ETTh1 | 96 | 0.382 | **0.399** | 0.386 | 0.405 | 0.423 | 0.448 | 0.479 | 0.464 | 0.384 | 0.402 | 0.386 | 0.400 | **0.376** | 0.419 | 0.449 | 0.459 |
| | 192 | **0.419** | **0.426** | 0.441 | 0.436 | 0.471 | 0.474 | 0.525 | 0.492 | 0.436 | 0.429 | 0.437 | 0.432 | 0.420 | 0.448 | 0.500 | 0.482 |
| | 336 | **0.474** | **0.445** | 0.487 | 0.458 | 0.570 | 0.546 | 0.565 | 0.515 | 0.491 | 0.469 | 0.481 | 0.459 | 0.459 | 0.465 | 0.521 | 0.496 |
| | 720 | **0.488** | **0.478** | 0.503 | 0.491 | 0.653 | 0.621 | 0.594 | 0.558 | 0.521 | 0.500 | 0.519 | 0.516 | 0.506 | 0.507 | 0.514 | 0.512 |
| | Avg | 0.441 | **0.437** | 0.454 | 0.447 | 0.529 | 0.522 | 0.541 | 0.507 | 0.458 | 0.450 | 0.456 | 0.452 | **0.440** | 0.460 | 0.496 | 0.487 |
| ETTh2 | 96 | **0.294** | **0.343** | 0.297 | 0.349 | 0.745 | 0.584 | 0.400 | 0.440 | 0.340 | 0.374 | 0.333 | 0.387 | 0.358 | 0.397 | 0.346 | 0.388 |
| | 192 | **0.377** | **0.395** | 0.380 | 0.400 | 0.877 | 0.656 | 0.528 | 0.509 | 0.402 | 0.414 | 0.477 | 0.476 | 0.429 | 0.439 | 0.456 | 0.452 |
| | 336 | **0.424** | **0.429** | 0.428 | 0.432 | 1.043 | 0.731 | 0.643 | 0.571 | 0.452 | 0.452 | 0.594 | 0.541 | 0.496 | 0.487 | 0.482 | 0.486 |
| | 720 | **0.422** | **0.437** | 0.427 | 0.445 | 1.104 | 0.763 | 0.874 | 0.679 | 0.462 | 0.468 | 0.831 | 0.657 | 0.463 | 0.474 | 0.515 | 0.511 |
| | Avg | **0.379** | **0.401** | 0.383 | 0.407 | 0.942 | 0.684 | 0.611 | 0.550 | 0.414 | 0.427 | 0.559 | 0.515 | 0.437 | 0.449 | 0.450 | 0.459 |
| Exchange | 96 | **0.083** | **0.201** | 0.086 | 0.206 | 0.256 | 0.367 | 0.094 | 0.218 | 0.107 | 0.234 | 0.088 | 0.218 | 0.148 | 0.278 | 0.197 | 0.323 |
| | 192 | **0.173** | **0.295** | 0.177 | 0.299 | 0.470 | 0.509 | 0.184 | 0.307 | 0.226 | 0.344 | 0.176 | 0.315 | 0.271 | 0.315 | 0.300 | 0.369 |
| | 336 | 0.326 | **0.412** | 0.331 | 0.417 | 1.268 | 0.883 | 0.349 | 0.431 | 0.367 | 0.448 | 0.313 | 0.427 | 0.460 | 0.427 | 0.509 | 0.524 |
| | 720 | 0.842 | **0.688** | 0.847 | 0.691 | 1.767 | 1.068 | 0.852 | 0.698 | 0.964 | 0.746 | 0.839 | 0.695 | 1.195 | 0.695 | 1.447 | 0.941 |
| | Avg | 0.356 | **0.399** | 0.360 | 0.403 | 0.940 | 0.707 | 0.370 | 0.413 | 0.416 | 0.443 | 0.354 | 0.414 | 0.519 | 0.429 | 0.613 | 0.539 |
| Weather | 96 | 0.167 | **0.205** | 0.174 | 0.214 | **0.158** | 0.230 | 0.202 | 0.261 | 0.172 | 0.220 | 0.196 | 0.255 | 0.217 | 0.296 | 0.266 | 0.336 |
| | 192 | 0.215 | **0.243** | 0.221 | 0.254 | **0.206** | 0.277 | 0.242 | 0.298 | 0.219 | 0.261 | 0.237 | 0.296 | 0.276 | 0.336 | 0.307 | 0.367 |
| | 336 | **0.269** | **0.285** | 0.278 | 0.296 | 0.272 | 0.335 | 0.287 | 0.335 | 0.280 | 0.306 | 0.283 | 0.335 | 0.339 | 0.380 | 0.359 | 0.395 |
| | 720 | **0.345** | **0.340** | 0.358 | 0.349 | 0.398 | 0.418 | 0.351 | 0.386 | 0.365 | 0.359 | 0.345 | 0.381 | 0.403 | 0.428 | 0.419 | 0.428 |
| | Avg | **0.249** | **0.268** | 0.258 | 0.279 | 0.259 | 0.315 | 0.271 | 0.320 | 0.259 | 0.287 | 0.265 | 0.317 | 0.309 | 0.360 | 0.338 | 0.382 |

## 5.4 Remaining Independence

According to the analysis in Section 4.1, there may not be any variable $V_j$ in the Markov boundary of $V_i$ that satisfies the conditional independence $V_i \perp\!\!\!\perp V_j | V \setminus V_j$, but there may still be unused independence conditions such as in Figure 1, $V_i \perp\!\!\!\perp V_s$ but $V_i \not\!\perp\!\!\!\perp V_s | V_c$. Thus, it follows that $I(V_i; V_p \cup V_c) < I(V_i; V_p \cup V_c \cup V_s)$, indicating that neglecting $V_s$ leads to information loss.

After the above process, we get the estimate function $\hat{f}$ to predict future sequence $X_{T+1:T+S} = \{x^1_{T+1:T+S}, x^2_{T+1:T+S}, \ldots, x^D_{T+1:T+S}\}$ from the historical sequence $X_{1:T} = \{x^1_{1:T}, x^2_{1:T}, \ldots, x^D_{1:T}\}$. To meet the ZCE constraint in Eq 6, for any variable $V_i$, we subtract the expectation given $S^i_2$ from the predicted results of $\hat{f}$ to get $M\hat{f}$ in Eq 9.

## 6 Experiment

In this section, we first provide the details of the implementation (Subsection 6.1). Then, we present the comparison results on six benchmark datasets (Subsection 6.2). Next, we conduct ablation studies to evaluate the effectiveness of each module in our method (Subsection 6.3).

### 6.1 Implement Details

All the experiments are implemented in PyTorch Paszke et al. (2019) and trained on NVIDIA V100 32GB GPUs. For the model architecture, we use ADAM Kingma & Ba (2015) with an initial learning rate in $\{10^{-3}, 10^{-4}\}$ and MSELoss for model optimization. An early stopping counter is employed to stop the training process after three epochs if no loss degradation on the valid set is observed. The mean square error (MSE) and mean absolute error (MAE) are used as metrics. All

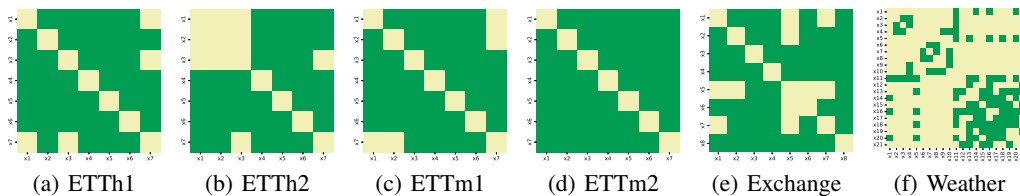

Figure 3: Visualization of the Markov boundaries for variables in benchmark datasets: ETTh1, ETTh2, ETTm1, ETTm2, Exchange, and Weather. Each row represents a specific variable, and green blocks indicate that the variable corresponding to the column is included in the Markov boundary of the variable represented by the row.

experiments are repeated 3 times and the mean of the metrics is used in the final results. The batch size is set to 4 and the number of training epochs is set to 10.

## 6.2 COMPARISON RESULTS

We thoroughly evaluate the proposed CAIFormer on various long-term time series forecasting benchmarks. For better comparison, we follow the experiment settings of iTransformer in (Liu et al., 2023b) the prediction lengths for both training and evaluation vary within the set $S \in \{96, 192, 336, 720\}$, with a fixed lookback length of $T = 96$.

We carefully choose 7 well-acknowledged forecasting models as our benchmark, including (1) Transformer-based methods: iTransformer Liu et al. (2023b), Autoformer Wu et al. (2021), FEDformer Zhou et al. (2022), Crossformer Zhang & Yan (2023); (2) Linear-based methods: DLinear Zeng et al. (2023), TiDE Das et al. (2023); and (3) TCN-based methods: TimesNet Wu et al. (2023).

Table 1 presents the results of CAIFormer in long-term multivariate forecasting with the best in **bold** and the second underlined. The lower MSE/MAE indicates the more accurate prediction result. Compared with iTransformer (Liu et al., 2023b), which uses variable attention, we improve in all datasets for different metrics.

## 6.3 ABLATION STUDY

In this section, we compare the performance of two causal discovery algorithms PC and FCI, and visualize the DAGs they generate on the ETTh1 dataset. Additionally, we validate the effectiveness of the mask discussed in Section 5.3 and the constraint mechanism introduced in Section 5.4.

### 6.3.1 CAUSAL DISCOVERY ALGORITHM

In this section, we choose two causal discovery algorithms for comparison, including (1) PC algorithm, which is a causal discovery method based on constraints such as conditional independence. It determines causal relationships by examining the dependencies between variables, the details ars prevet in Appendix B; (2) FCI, which handles potential hidden variables and circular causality through multiple conditional independence tests based on the PC algorithm. In appendix D, we visualize the DAG in ETTh1 dataset discovered by PC and FCI.

### 6.3.2 THE EFFECTIVE OF COMPONENTS

Following the setup in Section 6.2, we applied variable attention within the Transformer model to forecast Weather and ETTh1 datasets. To evaluate performance, we set both the Variable Attention Mask applied to the Transformer (discussed in Section 5.3) and the constraint-based collider structures within the Markov boundary (discussed in Section 5.4) optional, comparing their effects under different configurations. The lookback length $T = 96$ and prediction lengths $S \in \{96, 192, 336, 720\}$, the average prediction MSE and MAE for each dataset are shown in Table 2. The application of the variable Attention mask leads to improved predictive performance, indicating that the mask successfully prevents the model from considering correlations between irrelevant

Table 2: The average performance of lookback length $T = 96$ and prediction lengths $S \in \{96, 192, 336, 720\}$ in weather and ETTh1 datasets with variable attention Transformer.

| Variables Attention Mask | Collider Constrain | weather | | ETTh1 | |
|:---:|:---:|:---:|:---:|:---:|:---:|
| | | MSE | MAE | MSE | MAE |
| w/o | w/o | 0.258 | 0.279 | 0.454 | 0.447 |
| w | w/o | 0.251 | 0.272 | 0.445 | 0.440 |
| w | w | 0.249 | 0.268 | 0.441 | 0.437 |

variables. Similarly, constraining the hypothesis space using colliders from the Markov boundary enhances prediction accuracy, further validating the effectiveness of this constraint.

### 6.3.3 VISUALIZE VARIABLE ATTENTION MASK

For clarity, Figure 3 illustrates the Markov boundaries between variables from common multivariate time series forecasting datasets, as discussed in Section 5.2. In the figure, green blocks highlight the Markov boundary of the variable in the current row. The visualization reveals that while some variables are dependent, not all are interconnected. Additionally, Appendix B provides visualizations of the DAGs for these datasets.

## 7 CONCLUSION

In this paper, we introduce a novel causality-based algorithm, CAusal Informed Transformer (CAIFormer), to improve generalization in multivariate time series forecasting (MTSF) tasks. By leveraging causal discovery techniques, we construct a Directed Acyclic Graph (DAG) among variables and derive the Markov boundary to guide the model's attention mechanism. Our theoretical analysis shows that the Markov boundary, especially its collider structures, provides critical conditional independencies that can constrain the hypothesis space and reduce generalization error. Empirical evaluations on benchmark datasets demonstrate the advantages of CAIFormer.

### REPRODUCIBILITY STATEMENT

The theoretical results of this work are supported by well-defined assumptions, with complete proofs included in the appendix. Additionally, the algorithm's source code has been submitted as part of the supplementary materials.

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

APPENDIX

Appendix A presents the proof of Theorem 1. Appendix B details the implementation and results of the PC causal discovery algorithm. Appendix C provides an overview of the datasets used in this study. Appendix D offers an in-depth explanation of both the PC and FCI algorithms.

## A    PROOF OF THEOREM 1

The conditional expectation $\Pi : Z \in L^2(\Omega) \mapsto \mathbb{E}[Z|V_s]$ defines an orthogonal projection onto the space of $V_s$-measurable random variables with finite variance $L^2(\Omega, \sigma(V_s), P)$. Thus, its range and null space are orthogonal in $L^2(\Omega)$.

Let $f \in L^2(V)$. We have $Ef(V) = \mathbb{E}[f(V)|V_s] = \Pi f(V)$ hence $Ef(V)$ is in the range of $\Pi$. On the other hand,

$$\mathbb{E}[Mf(V)|V_s] = \mathbb{E}[f(V)|V_s] - \mathbb{E}[Ef(V)|V_s] = \mathbb{E}[f(V)|V_s] - \mathbb{E}[f(V)|V_s] = 0. \tag{19}$$

Therefore $Mf(V)$ is in the null space of $\Pi$. Finally, because $V_i \perp\!\!\!\perp V_s$ we have $\mathbb{E}[V_i|V_s] = \mathbb{E}[V_i] = 0$ by assumption, therefore $V_i$ is also in the null space of $\Pi$.

Hence, adopting this random variable view, the desired result simply follows from $L^2(\Omega)$ orthogonality:

$$\Delta(f, Mf) = \mathbb{E}[(V_i - f(V))^2] - \mathbb{E}[(V_i - Mf(V))^2] \tag{20}$$

$$= \|V_i - f(V)\|_{L^2(\Omega)}^2 - \|V_i - Mf(V)\|_{L^2(\Omega)}^2 \tag{21}$$

$$= \|V_i - Mf(V) - Ef(V)\|_{L^2(\Omega)}^2 - \|V_i - Mf(V)\|_{L^2(\Omega)}^2 \tag{22}$$

$$= \|V_i - Mf(V)\|_{L^2(\Omega)}^2 + \|Ef(V)\|_{L^2(\Omega)}^2 - \|V_i - Mf(V)\|_{L^2(\Omega)}^2 \tag{23}$$

$$= \mathbb{E}[Ef(V)^2] \tag{24}$$

$$= \|Ef\|_{L^2(\Omega)}^2. \tag{25}$$

## B    CAUSAL DISCOVERY VISUALIZATION

In this section, Algorithm 1 provides the pseudocode implementation of the PC algorithm, which includes three main steps: identifying the minimal set $S_{ab}$ that satisfies the conditional independence, directing edges, and finalizing the directed graph. We visualize the causal DAGs discovered by the PC algorithm across the ETTh1, ETTh2, ETTm1, ETTm2, Exchange, and Weather datasets in Figure 4. In these graphs, directed edges represent explicit causal relationships, while undirected edges denote uncertainty in causal direction.

## C    DATASET DESCRIPTIONS

In this paper, we conducted tests using eight real-world datasets. These datasets include: (1) ETT contains two sub-datasets: ETT1 and ETT2, collected from two electricity transformers at two stations. Each of them has two versions in different resolutions (15 minutes and 1h). ETT dataset contains multiple series of loads and one series of oil temperatures. (2) Weather covers 21 meteorological variables recorded at 10-minute intervals throughout the year 2020. The data was collected by the Max Planck Institute for Biogeochemistry's Weather Station, providing valuable meteorological insights. (3) Exchange-rate, which contains daily exchange rate data spanning from 1990 to 2016 for eight countries. It offers information on the currency exchange rates across different time periods.

We follow the same data processing and train-validation-test set split protocol used in iTransformer, where the train, validation, and test datasets are strictly divided according to chronological order to make sure there are no data leakage issues. The details of the datasets are provided in Table 3.

**Algorithm 1** Causal Discovery Algorithm-PC

**Input**: $\hat{P}$, a stable distribution on a set $V$ of variables;
**Output**: A pattern $H(\hat{P})$ compatible with $\hat{P}$.

1: **for** each pair of variables $a, b \in V$ **do**
2:     Search for a set $S_{ab}$ such that $(a \perp\!\!\!\perp b|S_{ab})$ holds in $\hat{P}$
3:     **if** no set $S_{ab}$ can be found **then**
4:         Connect vertices $a$ and $b$ with an edge in $G$
5:     **end if**
6: **end for**
7: **for** each pair of nonadjacent variables $a, b \in V$ with a common neighbor $c$ **do**
8:     **if** $c \notin S_{ab}$ **then**
9:         Add arrowheads pointing at $c$ (i.e., $a \rightarrow c \leftarrow b$)
10:     **end if**
11: **end for**
12: Orient as many of the undirected edges as possible in the partially directed graph
13: **while** there exists an undirected edge that can be oriented without creating a new v-structure or a directed cycle **do**
14:     Orient the edge
15: **end while**
16: **return** The directed graph as a pattern $H(\hat{P})$

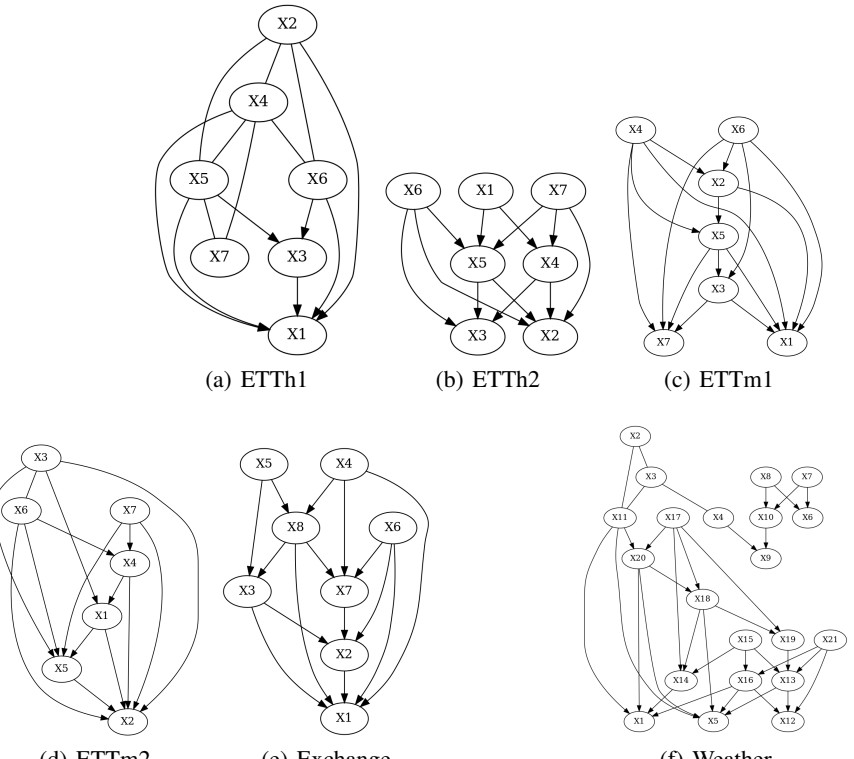

(a) ETTh1      (b) ETTh2      (c) ETTm1

(d) ETTm2      (e) Exchange      (f) Weather

Figure 4: Visualization DAG discovered by PC.

Table 3: Detailed dataset descriptions. *Dim* denotes the variate number of each data set. *Dataset Size* denotes the total number of time points in (Train, Validation, Test) split, respectively. *Prediction Length* denotes the future time points to be predicted, and four prediction settings are included in each data set. *Frequency* denotes the sampling interval of time points.

| Dataset | Dim | Prediction Length | Dataset Size | Frequency | Information |
|---|---|---|---|---|---|
| ETTh1,ETTh2 | 7 | {96, 192, 336, 720} | (8545, 2881, 2881) | Hourly | Electricity |
| ETTm1,ETTm2 | 7 | {96, 192, 336, 720} | (34465, 11521, 11521) | 15min | Electricity |
| Exchange | 8 | {96, 192, 336, 720} | (5120, 665, 1422) | Daily | Economy |
| Weather | 21 | {96, 192, 336, 720} | (36792, 5271, 10540) | 10min | Weather |

## D COMPARE PC WITH FCI

In the ablation study, we illustrate the differences between the PC and FCI algorithms. Figure 5 visualizes the DAGs discovered by both algorithms on the ETTh1 dataset. In the left figure, directed edges represent explicit causal relationships, while undirected edges indicate the absence of a fixed causal direction. In the right figure, $V_i \rightarrow V_j$ signifies that $V_i$ causes $V_j$, $V_i \circ\!\!\rightarrow V_j$ indicates that $V_i$ is not an ancestor of $V_j$, $V_i \circ\!\!-\!\!\circ V_j$ means no set $d$-separates $V_i$ and $V_j$, and $V_i \leftrightarrow V_j$ denotes the existence of a latent common cause between $V_i$ and $V_j$.

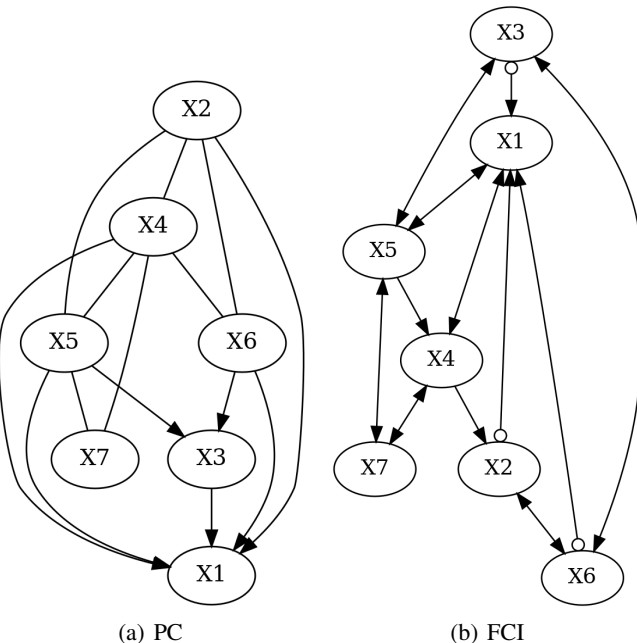

(a) PC                    (b) FCI

Figure 5: Visualization comparison PC with FCI.

