# OpenReview forum: "What Makes a Good Time-series Forecasting Model? A Causal Perspective"
_ICLR.cc/2025/Conference — ICLR 2025 Conference Withdrawn Submission_

### Official Review · Reviewer_LP4L · 2024-10-18

**Soundness:** 3
**Presentation:** 3
**Contribution:** 3
**Rating:** 6
**Confidence:** 4

**Summary:**

The paper aims to enhance the generalizability of the multivariate time-series forecasting task. It proposes a simple constraint from the collider structure within the Markov boundary and designs a novel CAIFormer model by restricting the attention module within the Markov boundary.

**Strengths:**

- The theoretical analysis looks good and solid. It gives an interesting constraint for improving the generalization error.
- The method is well-motivated. It is very simple and has the potential to incorporate many causal discovery methods.

**Weaknesses:**

- The theoretical results are not very clear to me. Please see the question part. Also, please add more background information, like how you define the generalization error.
- Can the method be applied to the classification setting?
- Do you make some assumptions on the distribution of the time series? Like no time-lag effect, the conditional distribution of V_i given its parents is invariant.
-  Would the method still have advantages when $v_c | (v_i,v_s)$ is changing?

**Questions:**

- in line 233, can you further clarify how the equation (4) is derived?
- Some notations are conflicts. In line 165, it is used to represent the edge set, and in line 149, it is used to represent an operator. Consider using another notation to represent the edge set.
- what is V used in section 4.2? If it is the node set, why only V_c and V_s are specified? What happened to other nodes?
- in equation 7, is E an operator that maps a function over (V_c, V_s) to a function on V_s?
- How do you define the generalization error? I am not sure how equation 10 would benefit the test performance.
- There are many expectations without specifying the distribution used.
- What is the motivation for choosing the L^2(V) space? Also, can you provide more background and interpretation about it?

---

### Official Review · Reviewer_9xye · 2024-10-29

**Soundness:** 2
**Presentation:** 2
**Contribution:** 2
**Rating:** 3
**Confidence:** 3

**Summary:**

- The authors explore the role of causal relationships in enhancing the generalization of multivariate time series models (MTSF - Multivariate time series forecasting); Graphical models can be used to narrow-down the space and improve generalization;
- The authors proposed Causal Informed Transformer (CAIFormer), which is a combination of:
1) Causal Discovery component to create DAG (PC-algorithm and FCI adopted);
2) Forecasting model with markov boundary informed by DAG; The model adopted was a transformer, where each variable attention module only focus on markov boundary;
- Author’s claim that adding all features might not be the best strategy; instead, explicitly considering these relationships can improve generalization performance
- As a first step, the authors want to identify max set of linear independent variables (Markov boundary); they examine the impact of collider structures on MTSF and show that enforcing these conditional indepen. can narrow down the hypothesis space.
- The last sessions are dedicated to an empirical evaluation;

**Strengths:**

- One positive point of the paper was the ablation testing the importance of subcomponents (Section 6.3.2).

- Authors also did a good job explaining the intuition.

**Weaknesses:**

Despite the good explanation of the intuition, I think there were some areas over simplified (e.g., Section 4.2):
- how much does these method scale? Causal discovery problems often struggle with dozen of variables; Is this a limitation to training the transformers? is this a limitation on the applications (as the authors mentioned practical applicability ).
- What if the relationship between variables is not linear? Is it realistic to always make this assumption?
- The formal definition of Markov boundary claims that a set of variables is a markov boundary if the markov blanket is minimal, meaning no variable can be dropped without losing information. Hence, if the authors can find a new variable that can be dropped (based on colliders), isn’t that a contraction that the previous set was markov boundary?
- I appreciate the empirical evaluation, but there is some information missing that would help demonstrate the superiority of the proposed method;

**Questions:**

1) Please elaborate on the following: the formal definition of Markov boundary claims that a set of variables is a markov boundary if the markov blanket is minimal, meaning no variable can be dropped without losing information. Hence, if the authors can find a new variable that can be dropped (based on colliders), isn’t that a contraction that the previous set was markov boundary?

2) Were the causal discovery methods used evaluated on how close they were from the truth graph? How robust are they to misspecified DAGs? How robust is the proposed method markov equivalent graphs?

3) Markov blankets / boundary are well-known to be sufficient and necessary for predictions (and by extension, forecasting) tasks. So when you say “ We explore the causal relationships among variables and discover that the Markov boundary is the sufficient and necessary subset of all variables in forecasting tasks” in the summary of contributions, what exactly are you adding?

4) In the empirical evaluation:

4.a) Did you re-run the baselines or adopt their reported values? If reported, can you add the citations on the table? if re-run, was the training regiment the same as the proposed method?

4.b) Could you run more than 3 seeds and report a confidence interval? Or if it is too expensive, is it possible to add error-bars?

4.c) Can you report how many parameters each method/baseline used?

4.d) What is the conclusion of 6.3.1 and how does that validate your method? The differences between PC and FCI are well-known, so I’m wondering about their interaction with the proposed method and what they add to the empirical evaluation of your proposed method.

---

### Official Review · Reviewer_ow1y · 2024-10-31

**Soundness:** 1
**Presentation:** 1
**Contribution:** 1
**Rating:** 3
**Confidence:** 3

**Summary:**

This paper introduces the role of causal relationships in enhancing the generalization of multivariate time series models. The Markov boundary of the DAG obtained from conditional independence constraints is applied as a mask to act on the attention scores of the Transformer to narrow the hypothesis space.

**Strengths:**

1. The paper utilizes causal models and conditional independence constraints to narrow down the hypothesis space, thereby enhancing the model's generalization ability in multivariate time series forecasting.
2. The paper presents a causality-based multivariate time series forecasting algorithm, CAusal Informed Transformer (CAIFormer), for time series forecasting.

**Weaknesses:**

**Main argument**

The role of causal relationships among variables in multivariate time series has not been well proved in this article. The obtained Markov boundary has been simply and rigidly used as mask mechanisms for attention scores. Although in section 4.2, it has been roughly proved that the hypothesis space can be constrained through conditional independence relationships to achieve better generalization, this still belongs to variable selection on the level of correlation relationships. Except for the DAG obtained by using the PC algorithm, the integration of the model with causality is not tight. The trained attention is considered to be able to learn which tokens are more worthy of attention and which ones need to be ignored. This study does not bring any new insights.

The experimental results cannot well prove that the method has a significant improvement, and some details are missing. For example, the model scale of CAIFormer is not mentioned in the paper. The content of the experimental part is insufficient, and it is difficult to draw conclusions.

There are many missing details:

1. It is important to express the existence of "collider provides additional independence relationships" before proposition 1. However, in proposition 1, when the condition set is given, Y is unexpectedly independent of X_i. Is this a conflict?
2. The differences in the attention matrix under different operations need to be considered (the original trained attention and the masked attention).
3. Why are the experimental results of this article a subset of the experimental results of iTransformer? The experimental results in different environments must be inconsistent. The experimental results are not rigorous.
4. The names of the variables in DAG are not given, lacking reliability to some extent.
5. the Markov boundary obtained according to the DAG does not consider its own variables, but from an autoregressive perspective it is important.
6. The difference in the ablation experiment effect is not significant, and more detailed results are needed in the experiment. For example, the average results of different prediction lengths should be shown separately.

**The paper has many imprecise parts, here are a few:**

1. For the experiment, a more detailed description of the settings is required.
2. The proof in section 4.2 needs to be more detailed.
3. There is a lack of comparison curves between the actuals (ground truth) and the predictions, and the performance cannot be intuitively presented.
4. What is the purpose of using PC and FCI for causal graph learning in the appendix? Does it help with the results of the paper?

**Things to improve the paper that did not impact the score:**

1. The experimental analysis needs to be more detailed.
2. The operator should be Hadamard product between the DAG and Attention, instead of add in figure 2.

**Questions:**

See the weaknesses above.

---

### Official Review · Reviewer_xUT6 · 2024-11-05

**Soundness:** 2
**Presentation:** 2
**Contribution:** 2
**Rating:** 3
**Confidence:** 3

**Summary:**

The submission presents a method that integrate causal knowledge, in the form of Markov blanket, with the aim of improving multivariate time-series forecasting.
The proposed approach combines causal discovery techniques with a transformer model.
The paper investigates theretical reasons why restricting the inputs to the markov Blanket.

**Strengths:**

- interesting problem
- some theoretical analysis
- simulation experiments
- mostly clear writing

**Weaknesses:**

1. while the paper deals with multivariate-time series forecasting, the causal framework presented and used is the one for iid data. Additionally the used algorithms, PC and FCI are algorithms for iid data mainly (while they can be adapted to time-series, by taking lag variables). The section on causality describes tools that are mainly used for iid data. For time series, they can be expanded but additional frameworks exist.  For instance, the causal graph for a time-series problem can obviously be cyclic since feedback loops can appear.
Under some simplifying assumptions, DAG and SEM  can be used to describe causal graphs in a time-series setting by unwrapping and explicitly representing lagged variables.
I feel that a proper discussion of causal discovery methods in time series settings is missing.
Some references:
 - The book "Elements of causal inference: foundations and learning algorithm" by Peters, Janzing, and Schölkopf. has a section on time series extension.
- "Discovering causal relations and equations from data" Camps-Valls et al. 2023 contains a review of time series causal discovery methods.

2. Also, a recently published paper (Neurips) https://arxiv.org/abs/2402.09891 tackles a similar problem, reaching a different conclusion.
   I understand that the theoretical argument is that causal information should help generalization and this is somehow a known fact. But how much this is useful in practice, especially with non-linear systems, is debatable. The only known method that tackles generalization that i know is anchor regression which is based on linearity assumptions. https://arxiv.org/abs/1801.06229.

3. the experiment section is somehow limited and the results are missing uncertainty.

**Questions:**

1.  what is a "maximal linearly independent group" ?
2. How do the results here relate to https://arxiv.org/abs/2402.09891?
3. How this method relates for example to anchor regression https://arxiv.org/abs/1801.06229 ?
4. Do you assume no instantaneous relationships ? if so DAG obtained with PC should be all directed  since the direction of time would impose a structure.
5. Have you tried classical or known time-series specific causal discovery approaches? e.g. Granger causality and non-linear extensions ? PCMCI and variants? tsFCI?

---

### Note · Authors · 2024-11-13

I have read and agree with the venue's withdrawal policy on behalf of myself and my co-authors.